# CryoEM structures of Kv1.2 potassium channels, conducting and non-conducting

Yangyu Wu, Yangyang Yan, Youshan Yang, Shumin Bian[†], Alberto Rivetta, Ken Allen, Fred J Sigworth*

Department of Cellular and Molecular Physiology, Yale University School of Medicine, New Haven, United States

*For correspondence:
fred.sigworth@yale.edu

Present address: [†]Department of Environmental Health and Safety, Yale University, New Haven, United States

Competing interest: The authors declare that no competing interests exist.

## eLife assessment

This **important** manuscript presents several structures of the Kv1.2 voltage-gated potassium channel, based on state-of-the-art cryoEM techniques and algorithms. The authors present **solid** evidence for structures of an inactivating mutant of Kv1.2, DTX-bound Kv1.2 and of Kv1.2 in potassium-free solution (with presumably sodium ions bound within the selectivity filter). These structures advance our knowledge of the molecular basis of the slow inactivation process of potassium channels.

**Abstract** We present near-atomic-resolution cryoEM structures of the mammalian voltage-gated potassium channel Kv1.2 in open, C-type inactivated, toxin-blocked and sodium-bound states at 3.2 Å, 2.5 Å, 3.2 Å, and 2.9 Å. These structures, all obtained at nominally zero membrane potential in detergent micelles, reveal distinct ion-occupancy patterns in the selectivity filter. The first two structures are very similar to those reported in the related Shaker channel and the much-studied Kv1.2–2.1 chimeric channel. On the other hand, two new structures show unexpected patterns of ion occupancy. First, the toxin α-Dendrotoxin, like Charybdotoxin, is seen to attach to the negatively-charged channel outer mouth, and a lysine residue penetrates into the selectivity filter, with the terminal amine coordinated by carbonyls, partially disrupting the outermost ion-binding site. In the remainder of the filter two densities of bound ions are observed, rather than three as observed with other toxin-blocked Kv channels. Second, a structure of Kv1.2 in $Na^+$ solution does not show collapse or destabilization of the selectivity filter, but instead shows an intact selectivity filter with ion density in each binding site. We also attempted to image the C-type inactivated Kv1.2 W366F channel in $Na^+$ solution, but the protein conformation was seen to be highly variable and only a low-resolution structure could be obtained. These findings present new insights into the stability of the selectivity filter and the mechanism of toxin block of this intensively studied, voltage-gated potassium channel.

## Introduction

Among the most-studied ion channel proteins is the Kv1 family of tetrameric, six-transmembrane-segment, voltage-gated potassium (Kv) channels. The best-known member of this family is the *Drosophila* Shaker channel, first cloned in the late 1980s (*Iverson et al., 1988*; *Papazian et al., 1987*) and which quickly became the favorite object for biophysical studies due to its small size and ease of heterologous expression. The potassium current in the squid giant axon described by *Hodgkin and Huxley, 1952* is carried by a closely related Shaker homologue (*Rosenthal and Bezanilla, 2002*).

In 2005, *Long et al., 2005* obtained the 2.9 Å crystal structure of Kv1.2, a mammalian Shaker-family channel. This structure showed clearly the overall architecture of the channel, but unfortunately

the densities were weak in the all-important voltage-sensor domains (VSDs). Two years later, *Long et al., 2007* presented the 2.4 Å X-ray structure of the 'paddle chimera' Kv1.2–2.1 channel in which a portion of the voltage-sensor domain (the S3-S4 paddle) was replaced by the corresponding Kv2.1 sequence. This structure presented the VSDs in fine detail, and so was a watershed in the structural understanding of the many functional measurements of Shaker channels. It has also formed the basis of computational studies of voltage-sensing and gating mechanisms. A subsequent cryoEM structure of Kv1.2–2.1 channels reconstituted into nanodiscs (*Matthies et al., 2018*) has excellent agreement with the X-ray structure. Meanwhile, a very welcome recent development is the cryoEM structure of the original Shaker channel itself (*Tan et al., 2022*).

The high-resolution structure of the Kv1.2 paddle chimera channel has been the basis for much research, but the native Kv1.2 channel has been used for more functional studies (*Ishida et al., 2015*; *Suárez-Delgado et al., 2020*; *Wu et al., 2022*). The properties of the native and chimeric channels are similar but not identical (*Tao and MacKinnon, 2008*). The chimeric channel has 27 amino-acid differences in the S3-S4 paddle region, but the VSD structures of *Long et al., 2005*; *Long et al., 2007* are remarkably similar; in the transmembrane region the $\alpha$-carbon traces are nearly superimposable. Our structure of Kv1.2 confirms this remarkable structural conservation.

Inactivation is an important process that diminishes voltage-gated channel current even when the activating membrane depolarization is maintained. A relatively slow inactivation process, usually called C-type inactivation (*Hoshi et al., 1991*) involves conformational changes in the pore domain and the selectivity filter. Various mutations in the pore domain have been shown to accelerate or impede C-type inactivation. The Shaker pore-domain mutation W434F renders the channel almost completely non-conductive through a process like C-type inactivation (*Perozo et al., 1993*; *Pless et al., 2013*; *Suárez-Delgado et al., 2020*; *Yang et al., 1997*). Structures of this mutant (*Tan et al., 2022*) and of a corresponding mutant Kv1.2–2.1 paddle-chimera channel (*Reddi et al., 2022*) show a dramatic dilation of the selectivity filter that disrupts two of the four ion-binding sites in the selectivity filter. We report here the structure of the corresponding W366F mutant in the Kv1.2 channel background, which shows a nearly identical dilation.

An important aspect of C-type inactivation is that it is enhanced when potassium ions are absent. In KcsA the replacement of most $K^+$ with $Na^+$ results in a collapse of the selectivity filter region, occluding the two central ion-binding sites (*Zhou et al., 2001*). Other channels including NaK2K and members of the K2P family show a destabilization of the external part of the pore at low $K^+$ concentrations, abolishing the two outer ion-binding sites (*Lolicato et al., 2020*; *Matamoros et al., 2023*; *Sauer et al., 2011*). In low potassium, the NaK2K channel actually shows a dilation of the outer selectivity filter very similar to that seen in Shaker-W434F and the corresponding K1.2–2.1 mutant. On the other hand, a lengthy molecular-dynamics simulation of deactivation in the Kv1.2–2.1 chimera channel in symmetrical $K^+$ solutions showed that, with the closing of the channel gate, there is a complete loss of water and ions from the cavity below the selectivity filter. Under these conditions, however, the selectivity filter remains intact and is populated by $K^+$ ions (*Jensen et al., 2012*; M. Ø. Jensen, personal communication). In view of the variety of effects of low $K^+$, in our current study we sought to observe experimentally the structure of the Kv1.2 channel when potassium ions are absent.

Finally, Kv channels are targets for many toxins that inhibit channel conductance, with a common mechanism being occlusion of the pore. This occlusion has been seen directly in two cases. Crystal structures of the Kv1.2 paddle-chimera channel have been obtained with charybdotoxin (CTx) bound (*Banerjee et al., 2013*), and cryoEM structures of Kv1.3 with the ShK toxin bound (*Selvakumar et al., 2022*). In the present work, we consider dendrotoxins (DTxs) produced by the mamba snake *Delodiaspis*. These are distinct from CTx but also block Kv channels with high potency and selectivity (*Gasparini et al., 1998*). In this study, we have determined the structure of Kv1.2 with α-DTX bound.

## Results
### Overview of mammalian Kv1.2 structure

For this study, we employed a full-length Kv1.2 $\alpha$-subunit construct containing three mutations in disordered regions of the N-terminus and S1-S2 linker. We denote this construct used for structure determination Kv1.2$_s$. We co-expressed a construct of of Kvβ2 (*Gulbis et al., 1999*) containing residues 36–367. Each β2-subunit contained five mutations chosen to neutralize positive charges on the

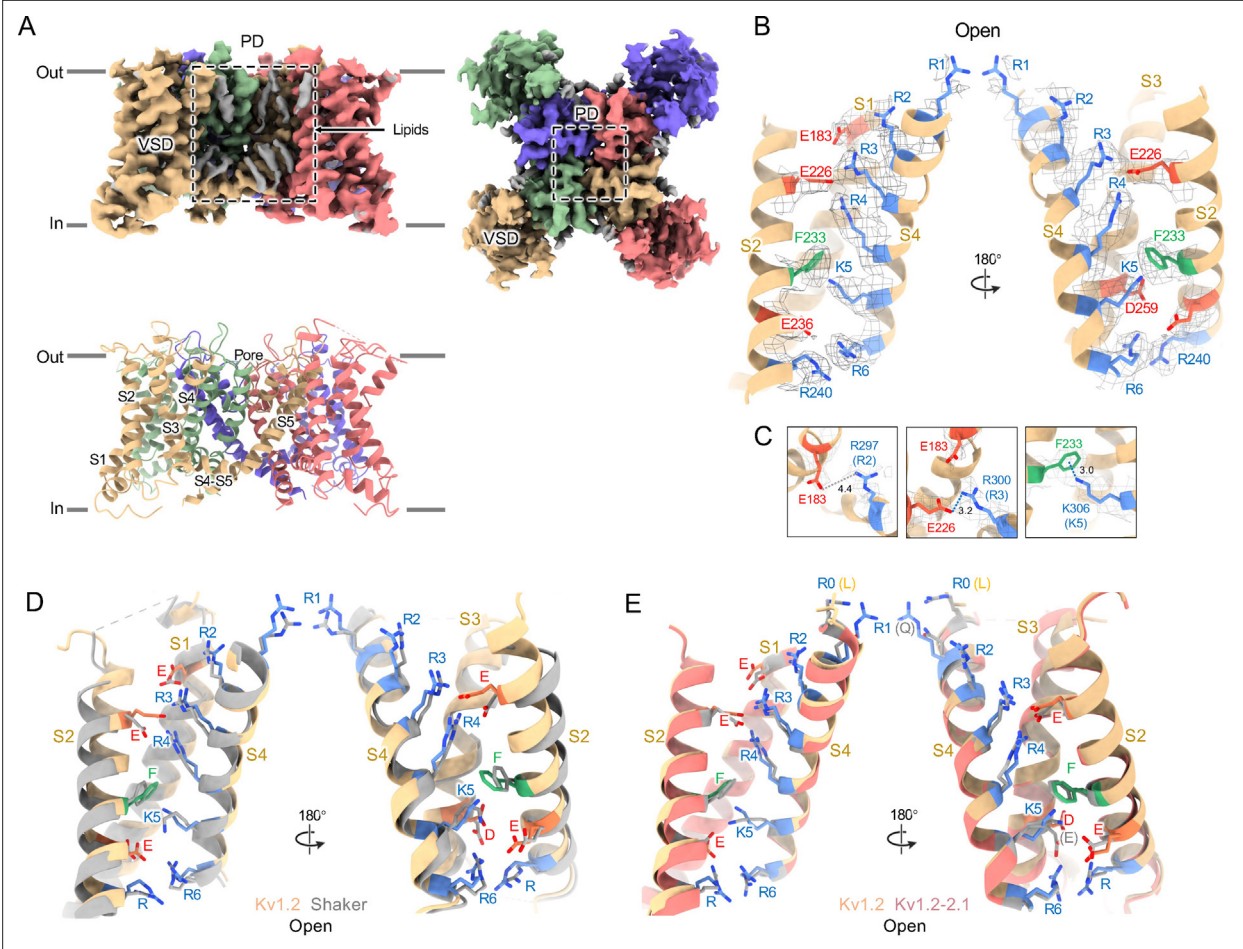

**Figure 1.** Open Kv1.2 overall structure. (**A**) Side and top view of the Kv1.2$_s$ cryoEM density map (upper panel) and model (lower panel). Lipid densities are colored gray in the map. (**B**) Side view of VSD structure with map density of Kv1.2$_s$. (**C**) The relative positions of the interacting residues R2 and E138 (upper), R3 and E226 (middle), K5 and F233 (lower) are shown. (**D**) Superposition showing the very close match of Kv1.2 (yellow) and Shaker (gray) VSD structures. (**E**) Superposition of Kv1.2$_s$ (yellow) and Kv1.2–2.1 (pink) VSD structures. Positively charged, negatively charged and aromatic residues are shown as blue, red and green, respectively. VSD, voltage-sensing domain; PD, Pore domain.

The online version of this article includes the following figure supplement(s) for figure 1:

**Figure supplement 1.** Image processing and reconstruction of Kv1.2$_s$.

cytoplasmic face of the subunit; these remove strong, interfering interactions with cryoEM carbon substrates. We find that the mutations have no effect on β-subunit secondary structure. Although α and β subunits were co-expressed, in cryoEM micrographs we observed many $α_4$ complexes along with the expected $α_4β_4$ particles. As the structures of the transmembrane regions of these two populations of channels were indistinguishable, we combined the two particle sets. This is reasonable as the presence of β2 subunits has very little effect on the structure of the α-subunit T1 domains to which they bind (*Gulbis et al., 2000*) and also little effect on channel function (*Rettig et al., 1994*).

The constructs encoding the Kv1.2$_s$ or Kv1.2$_s$-W366F mutant α-subunits, along with the β2 subunits, were expressed in *Pichia pastoris* essentially as described (*Long et al., 2005*). Channel complexes were affinity-purified in the presence of dodecylmaltoside detergent and subjected to size-exclusion chromatography in buffers containing either 150 mM K$^+$ or 150 mM Na$^+$ ions. The channel complexes were plunge-frozen on grids for cryoEM analysis. Focusing on the transmembrane region of the Kv1.2 channel complexes, we obtained structures of nominally open, C-type inactivated, DTx-bound, and K$^+$-free states at resolutions of 3.2 Å, 2.5 Å, 3.2 Å, and 2.9 Å, respectively. Most main-chain and side-chain densities were clearly visible in the resulting maps allowing atomic models to be built with high confidence.

Structures of the Kv1.2$_s$ channel (*Figure 1*, *Figure 1—figure supplement 1*) are very similar to those of the Kv1.2–2.1 chimera (*Long et al., 2007*) and *Drosophila* Shaker (*Tan et al., 2022*) channels. As observed in other Kv1 and Kv2 structures, the voltage-sensing domains (VSDs) contain the membrane-spanning S1-S4 helixes (*Figure 1D and E*) while the S5, S6, pore helix and selectivity filter P-loop form the ion-conduction pore in a domain-swapped configuration. Densities of lipids bound to the VSDs and pore domains (PDs) are clearly visible in the Kv1.2$_s$ maps (*Figure 1A*).

In the critical region of the voltage-sensor domain (VSD) the sidechains of the voltage-sensing Arg and the coordinating Glu and Asp residues, as well as the charge-transfer center phenylalanine (*Tao et al., 2010*) are essentially superimposable with Shaker (*Figure 1D*), with an RMS difference of all Arg sidechain atoms of 0.85 Å. As Kv1.2 is a member of the Shaker potassium channel family and has 68% amino-acid identity with *Drosophila* Shaker, it is not surprising that the fold is very similar.

As in the other homologues an open S6 gate is visible in the detergent-solubilized Kv1.2$_s$ structure; this is expected for the open conformation at zero membrane potential. Seeing no evidence of a barrier to ion flux, we call this the open-channel structure (*Figure 2B*).

Despite the very closely matched open-state structures of the VSDs it is therefore puzzling that the total gating charge movement in Shaker channels is larger, about 13 elementary charges per channel, while it is only 10 $e_0$ per channel in Kv1.2 (*Islas, 2016*). The most likely explanation is that in Kv1.2 there is some additional restraint, lacking in Shaker, on the physical displacement of each S4 helix.

In the paddle chimera a region including the N-terminal half of the S4 helix in Kv1.2 is replaced by the corresponding Kv2.1 sequence. One difference in the voltage-sensing residues is that a glutamine replaces the arginine at the first position (R1) in the chimera. Another difference is that there is an earlier Arg residue R0 that replaces leucine, but is unlikely to contribute to charge movement. A direct measurement of charge movement in the paddle chimera is lacking, but macroscopic activation and inactivation show lower voltage sensitivity (*Tao and MacKinnon, 2008*) as if its gating charge movement were reduced. Nevertheless, as was observed by *Long et al., 2007*, the structures of the S4 regions are very similar: comparing Kv1.2 with the paddle chimera (*Figure 1E*) the rms deviation of Arg sidechain atoms is less than 0.75 Å.

## Channel inactivation

*Figure 2* compares the structure of the open channel with that of the Kv1.2$_s$-W366F mutant. While the corresponding mutation W434F in Shaker essentially abolishes ion current through the channel (*Yang et al., 1997*) the Kv1.2$_s$ mutant allows currents to flow transiently on depolarization, but decay to less than 3% of the peak within 80ms (*Figure 2—figure supplement 1*). Inactivation in these channels can be slowed by raising extracellular $K^+$ or by applying tetraethylammonium ion (*Suárez-Delgado et al., 2020*), properties that are hallmarks of C-type inactivation. Here, we shall call the cryoEM structure of Kv1.2$_s$-W366F the 'inactivated channel' structure, keeping in mind that other distinct inactivated channel conformations are likely to exist. The overall conformational difference between open and inactivated structures in Kv1.2s is identical to the difference observed in structures of Shaker (*Tan et al., 2022*) and in structures of Kv1.2–2.1 (*Long et al., 2007*; *Reddi et al., 2022*).

For detailed comparison with structures of other voltage-gated potassium channels, we borrow from *Miller, 1991* the numbering of the 39 residues in the S5-S6 linking region, which we will denote as residue numbers 1' to 39'. (*Figure 2A*). The 39-residue numbering is appropriate for Kv1 through Kv4 channels (*Shealy et al., 2003*) and is fortuitously convenient in comparisons with KcsA, because it differs numerically by exactly 50 from the residue numbering of KcsA. In the Miller system for example the Kv1.2 W366 residue is W17', and we can denote the W366F mutant channel as Kv1.2$_s$-W17'F. In KcsA the W17' residue is W67.

The selectivity filter of potassium channels consists of an array of four copies of the extended loop (the P-loop) formed by a highly conserved sequence, in this case TTVGYGD. The main-chain carbonyl oxygens of the residues TVGY, positions 25' to 28', delineate ion binding sites S3, S2 and S1, respectively, while the lower (most intracellular) edge of binding-site S4 is formed by the T25' hydroxyl. Hydrogen bonds involving two P-loop residues anchor the outer half of the selectivity filter and are particularly important in inactivation mechanisms (*Figure 2B*, right panels) (*Pless et al., 2013*; *Sauer et al., 2011*). Normally, the tyrosine Y28' (Y377 in Kv1.2) is constrained by hydrogen bonds to residues in the pore helix and helix S6 and is key to the conformation of the selectivity filter. The final aspartate

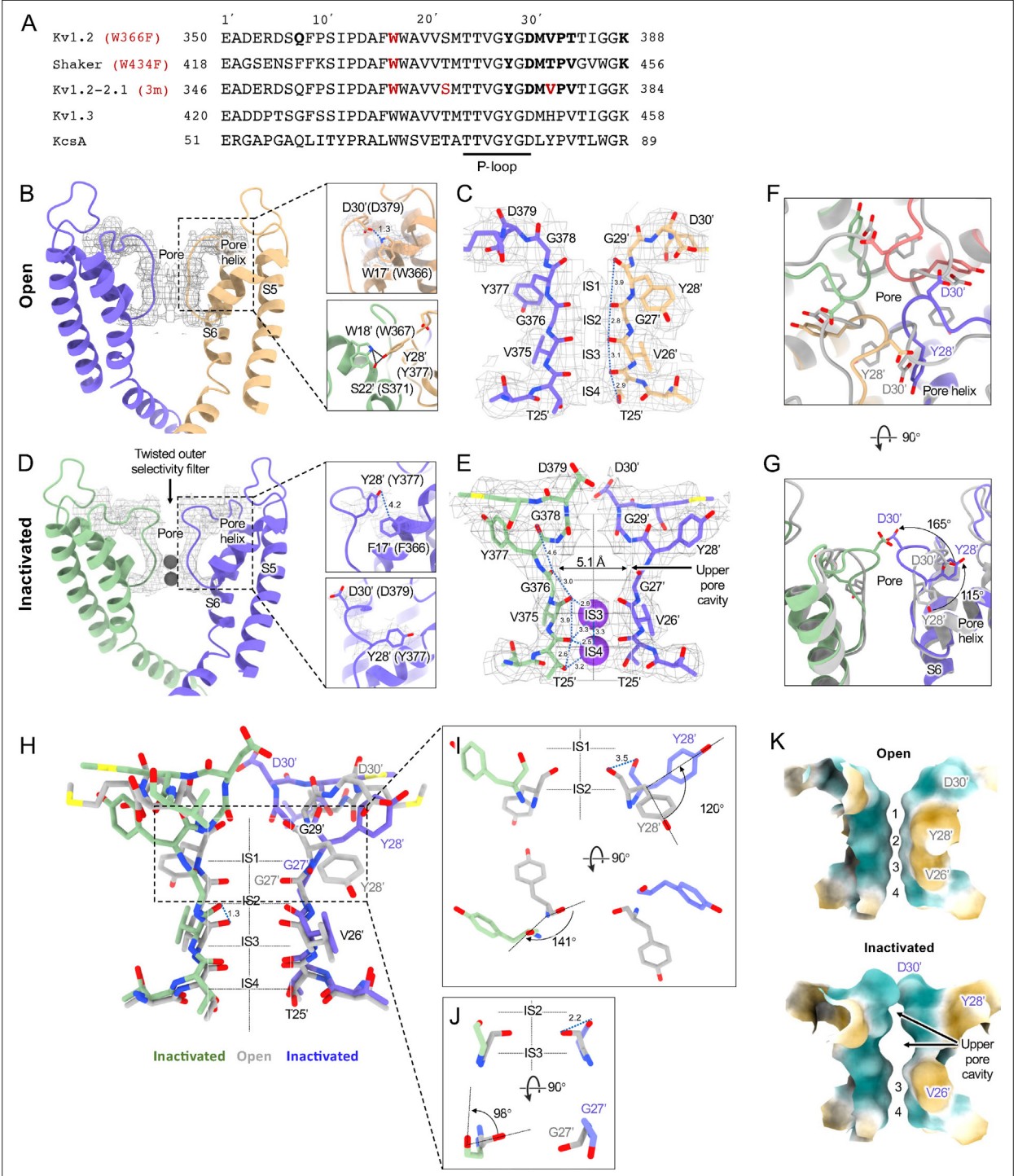

**Figure 2.** Kv1.2 pore domain and selectivity filter structures in inactivated and open conformations. (**A**) Sequence alignment of potassium channels in the S5-S6 linker region, with the relative numbering of *Miller, 1991* indicated at top. Locations of mutations of interest are noted in red. (**B**) Side view of opposing subunits in the open Kv1.2ₛ pore domain. Relative positions of Y28', W17' and D30' are shown in the right-hand panels. Shown with a dashed blue line is the key hydrogen bond between D30' and W17' that is eliminated in the W17'F (W366F) mutant. (**C**) Side view of opposing Kv1.2ₛ P-loops. Labels on the left show the Kv1.2 residue numbering, and on the right the Miller numbering. (**D**) Side view of the inactivated Kv1.2ₛ W17'F pore domain. The new locations of Y28', F17' and D30' are shown in the right panels. (**E**) Side view of the Kv1.2ₛ P-loop in the inactivated conformation. In the large upper-pore cavity the G27' and G29' carbonyls are 5.1 and 11 Å apart. Potassium ions are shown as purple balls. (**F**) Superimposed top views of the Kv1.2ₛ outer pore in inactivated (colored) and open (grey) states. In the inactivated channel the displaced Y28' ring of one subunit is in the position occupied by D30' – in the neighboring subunit – of the open channel. (**G**) The corresponding side view. The large rotation of the Y28' side chain and

*Figure 2 continued on next page*

*Figure 2 continued*

flipping of D30′ are indicated by curved arrows. (**H**) Side view of the P-loop in inactivated (colored) and open (grey) conformations. (**I,J**) Details of Y28′ G27′ carbonyl and side chain reorientation from open (grey) to inactivated (colored) states are shown in side view (upper) and top view. (**K**) Surface renderings of open and inactivated selectivity filter region. Hydrophilic and hydrophobic surfaces are shown in teal and orange, respectively.

The online version of this article includes the following figure supplement(s) for figure 2:

**Figure supplement 1.** Currents from native and W366F Kv1.2 channels.

**Figure supplement 2.** Processing of Kv1.2 W366F images.

**Figure supplement 3.** Voltage-sensing-domain conformational differences between open and C-type inactivated states.

**Figure supplement 4.** Stereo views of selectivity filter and voltage-sensing-domain.

---

of the P-loop, D30′ (D379 in Kv1.2) is normally located near the extracellular surface and has a side chain that also participates in H-bonds with W17′ (W366 in Kv1.2) on the pore helix.

The difference between the open and inactivated Kv1.2$_s$ structures, like the open-inactivated state differences in Kv1.2–2.1 (**Reddi et al., 2022**) and in Shaker (**Tan et al., 2022**) can be imagined as resulting from a two-step process. The first step is a partial twist of the P-loop backbone involving D30′. When W17′ is mutated to phenylalanine, it is no longer an H-bond donor to D30′ (**Figure 2D** right upper panel). The resulting destabilization of D30′ allows it to reorient toward the external water-filled vestibule.

The second step is the reorientation of Y28′ and further twisting of the polypeptide backbone. Y28′ normally participates in H-bonds with the pore-helix residues W18′ and S22′ (**Figure 2B**, right-hand panels), but with the release of D30′, and presumably the entry of water molecules into the space surrounding the P-loop, the side chain of Y28′ also reorients toward the external solution, filling some of the original volume occupied by the side chain of D30′. The reorientation of the phenol group of Y28′ is through a very large pitch angle of about 120° (**Figure 2G,H,I**). The reorientations of the D30′ and Y28′ side chains drag and twist the backbone. The result is an enlargement of the ion-binding site IS2 to a width of 5.1 Å between opposing carbonyl oxygens (**Figure 2E**) and an even wider (11 Å) cavity is formed at the level of IS1. There are no clear ion densities in the upper selectivity filter.

Meanwhile, the lower binding sites IS3 and IS4 show relatively small disturbances in the inactivated state. The largest change is a rise of 1.3 Å in the position of the G27′ carbonyl, the one which defines the top of IS3 (**Figure 2J**). As in the inactivated Shaker structure (**Tan et al., 2022**) strong ion densities are seen in IS3 and IS4.

Might symmetry-breaking accompany Kv1.2 inactivation? MD simulations of inactivated paddle chimera, KcsA and hERG channels (**Kondo et al., 2018**; **Li et al., 2021b**; **Li et al., 2021a**), all homo-tetramers, show a change to twofold symmetry in the pore region such that the selectivity filter has distinct conformations for each pair of opposing subunits. However, we were not able to detect any evidence of reduced symmetry in the inactivated Kv1.2 channel. We were able to build the atomic model unambiguously into the 2.5 Å map that was obtained with C4 symmetry imposed. Further, there was no evidence of broken symmetry when using symmetry expansion and local C1 refinements of the cryoEM dataset.

A comparison of our open-channel and inactivated Kv1.2$_s$ structures show subtle but notice-able differences in the VSDs. Salt bridges involving the S4 Arg and Lys residues are shifted slightly (**Figure 2—figure supplement 3A–D**). Arg300 (R3) is in close proximity to Glu226 on the S2 helix for the open channel, while R3 is closer to Glu183 in the S2 helix. The Glu226 side chain adopts a visible interaction with R4 in the inactivated state. In both open and inactivated states of Kv1.2$_s$, K5 interacts more closely with S2 Phe233 than is the case in the paddle chimera channel (**Figure 1E**).

Functional interactions have been observed between residues in the voltage-sensing domains and pore residues involved in C-type inactivation (**Bassetto et al., 2021**; **Conti et al., 2016**). Leucine and valine residues on the S4 helix interface with serine and leucine residues on S5, which in turn contact F16′ and W17′ on the pore helix. We see very little relative movement of the residues at the S4-S5 interface, much less than 1 Å difference between open and inactivated structures.

When individual VSD domains are aligned the VSD helices in Kv1.2$_s$ and the inactivated Kv1.2$_s$-W17′F superimpose very well at the top (including the S4-S5 interface described above), there is a general twist of the VSD helix bundle that yields an overall rotation of about 3° at the bottom of the bundle (**Figure 2—figure supplement 3E–F**). In moving to the inactivated state, the axes of the

helices S0, S1 and S2 tilt by 6°, 4°, and 3.2° in a clockwise direction as viewed from the cytoplasmic side, while S4 is stationary. This lower VSD rotation provides a good explanation of the shifts in the S2 residues Glu183 and Glu226 that interact with R2 and R3 (*Figure 2—figure supplement 3A–D*). In contrast to Kv1.2$_s$, neither Shaker and Kv1.2–2.1 structures show this rotation between open and inactivated states (*Figure 2—figure supplement 3G–H*) but instead the VSDs remain superimposable.

It should be noted that the difference between Kv1.2$_s$ and the others in conformational changes might arise from the variety of systems used for structure determination, which include DDM micelles, MSP-bounded nanodiscs, and crystals grown in different laboratories (but under very similar conditions and having the same space group). In the Kv1.2$_s$ case for example, the DDM detergent micelles might allow the VSDs to be particularly mobile.

## A dendrotoxin blocks Kv1.2 by inserting a lysine into the pore

Dendrotoxins are peptide neurotoxins from mamba snakes that bind with nanomolar affinities and block potassium channels. Alpha-dendrotoxin (α-DTX) consists of a peptide chain of 59 amino acids stabilized by three disulfide bridges (*Figure 3A*) and, like other dendrotoxins, exhibits the same fold as Kunitz protease inhibitors (*Skarzyński, 1992*). In α-DTX arginine and lysine residues are concentrated near the N-terminus (Arg3, Arg4, Lys5), the C-terminus (Arg54, Arg55) and at the narrow β-turn region (Lys28, Lys29, Lys30). The Lys5 side chain protrudes from the surface of the molecule, and its modification by acetylation markedly reduces binding (*Harvey et al., 1997*). Replacement of Lys5 by alanine or ornithine decreased the affinity for potassium channels by 1000-fold and 100-fold, respectively, suggesting that the sidechain geometry is critical (*Gasparini et al., 1998*). The Lys5 sidechain therefore has been the prime candidate for the pore-blocking moiety.

By adding an excess of α-DTx to detergent-solubilized Kv1.2 protein we obtained a dataset of about 300,000 particles where an additional cap of density can be seen outside the pore (*Figure 3B*). In 3D classification only a small fraction (<1%) of toxin-free particles was detected, and this class was removed before further processing. The processing of particle images is summarized in *Figure 3—figure supplement 1*. Refined structures with imposition of fourfold symmetry (C4) or with no symmetry (C1), yielded resolutions 2.8 and 3.2 Å, respectively. In the C4 reconstruction, the additional cap of density from the toxin was not interpretable, as it arises from the four-fold superposition of the asymmetric toxin bound in alternative poses to the symmetric channel. *Zhang et al., 2021* have demonstrated however that small asymmetric ligands can be distinguished in large single-particle datasets by focused classification and refinement. Similarly we used symmetry expansion, a reference mask covering only the toxin and upper selectivity filter, and a set of four asymmetric starting references to obtain, through 3D classification without alignment in the Relion software, a C1 reconstruction. The refined C1 density map and model are shown in *Figure 3C and D*. Due to the C4 symmetry of Kv1.2, there are four equivalent DTx binding sites on the channel. The top and side views display the DTx on the negatively charged outer mouth of the Kv1.2 channel pore, with the density map fitted to the atomic model (*Figure 3E*).

The X-ray structure of α-DTX (*Skarzyński, 1992*) could be docked with minor adjustments into the map, with the Lys5 sidechain extending into the channel pore. The positively-charged toxin is tethered to the negatively-charged outer mouth of the pore by three salt bridges: toxin residues K30, R15, and R3 bind to three channel residues D355, D363, and D363 (D6', D14', and D14' in the turret and pore helix). Each of the three channel residues is located on a different subunit (*Figure 3G and H*).

In the selectivity filter of the toxin-bound channel (*Figure 3I*), a continuous density is seen to extend downward from the toxin to the boundary of IS1. This density is well modeled by the side chain of Lys5, with the terminal amine coordinated by the carbonyls of Y377 (Y28'). We conclude that block by α-DTx is occlusion of the selectivity filter through binding of the Lys5 terminal amine. In the selectivity filter a large density peak, presumably a K$^+$ ion, is seen in a somewhat offset IS1 binding site; a second, larger K$^+$ ion peak is seen in IS3.

Like the nearby Lys5 residue, substitution of alanine for Leu9 also results in a 1000-fold reduction in toxin affinity (*Gasparini et al., 1998*). In the structure, however, the Leu9 sidechain does not show contacts with either the channel or other residues in the toxin itself. It is therefore not obvious why it is important in toxin binding.

We also tried mixing an excess of α-DTX with the Kv1.2s W366F protein sample in an attempt to observe a pore-blocker-bound inactivated structure. However, we found no evidence of bound toxin

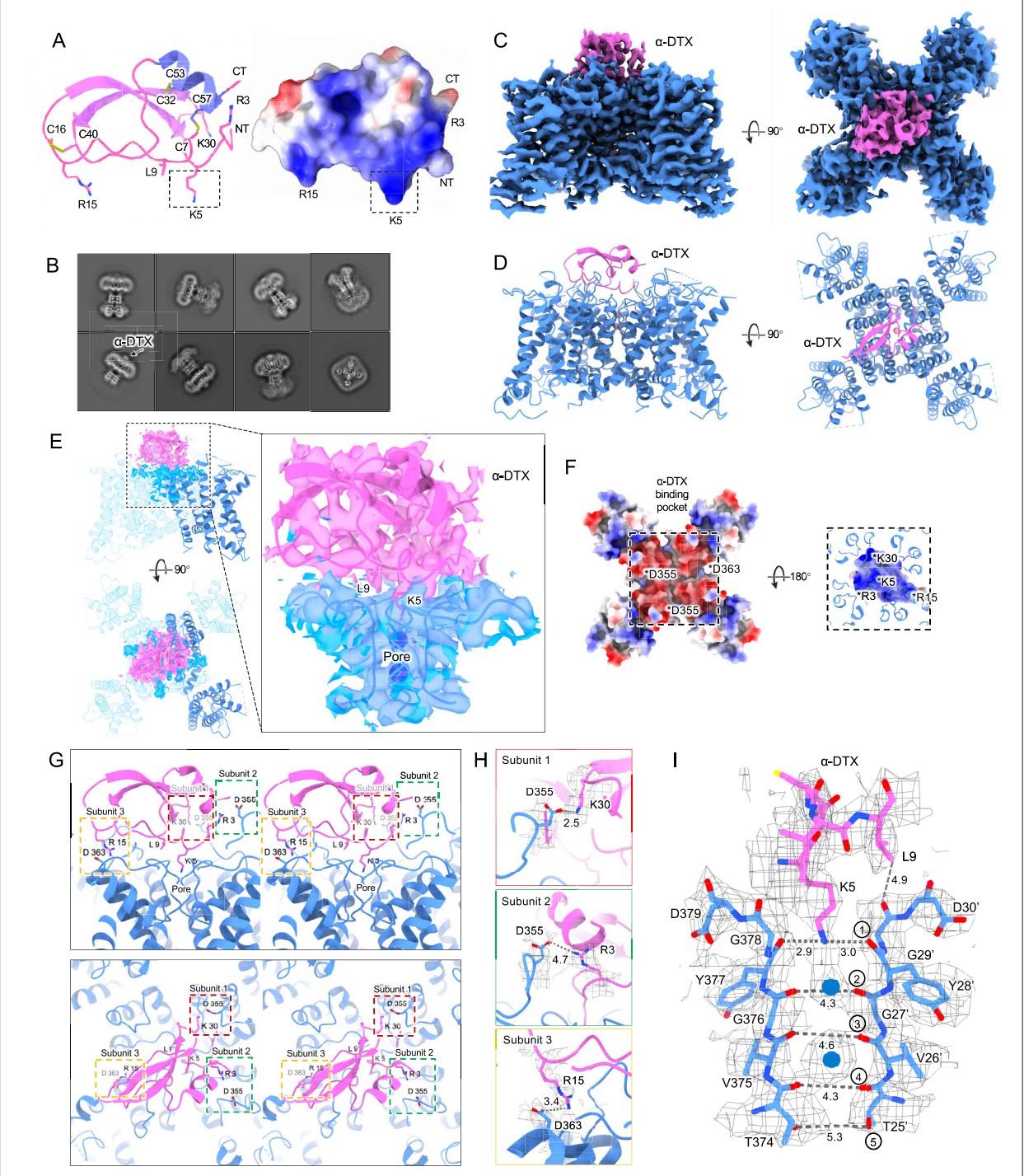

**Figure 3.** Kv1.2s DTx-bound structure. (**A**) DTx structure and electrostatic surface view. Basic residue sidechains are illustrated, and disulfide bridges are shown in yellow. (**B**) Representative 2D classes from the DTx-bound Kv1.2s. Box size is 273 Å. (**C**) Side and top view of Kv1.2-DTx cryoEM density map, obtained with no symmetry imposed. (**D**) Side and top view of the fitted model. (**E**) Interaction interface of DTx and Kv1.2s, illustrated with map density of the toxin and pore domain, with superimposed models including one complete subunit. (**F**) Top-down view of the Kv1.2s-DTx structure with DTx removed, showing the channel in electrostatic surface view (left panel); and the corresponding bottom-up view through the DTx-channel interface (right panel) where DTx is shown in electrostatic surface view, and extensions of Kv1.2s helices in blue. The asterisks indicate the charged groups. (**G**) Three salt-bridge interactions between DTX and Kv1.2s are shown as side (upper panel) and top (lower panel) stereo view. (**H**) Close-up view of the three salt-bridge interactions from the horizontal plane with cryoEM density map. (**I**) Side view of the selectivity filter of Kv1.2s-DTx shown with the toxin model for residues K5 to L9. Dashed lines show approximate distances between carbonyl oxygens. Potassium ions are shown as blue balls.

*Figure 3 continued on next page*

*Figure 3 continued*
The online version of this article includes the following figure supplement(s) for figure 3:

**Figure supplement 1.** CryoEM imaging and reconstruction of Kv1.2$_s$-DTX.

**Figure supplement 2.** Comparison of the Kv1.

density on the channel in the cryoEM 2D classes. This is no surprise in view of the large rearrangement of the extracellular pore entrance in the inactivated state, where the IS1 and IS2 sites are abolished and the corresponding carbonyls point away from the pore axis (*Figure 2E*), eliminating the site of binding of the lysine side chain. It should be noted however that in the case of a ShK toxin derivative binding to the Kv1.3 channel, *Tyagi et al., 2022* observed that binding occurs to inactivated channels. In that case the inactivated state has a less drastically remodeled selectivity filter, and binding of the toxin restores the selectivity filter to its conducting configuration (*Tyagi et al., 2022*).

## The Kv1.2 selectivity filter in nominally K$^+$-free medium

The X-ray crystal structure of the KcsA channel in low K$^+$ solution (*Zhou et al., 2001*) shows a collapsed selectivity filter, in which the residues V26' and G27' are rearranged, abolishing the binding sites IS2 and IS3. In low K$^+$ other channels including KirBac2.2 and K2p10.1 show large conformational fluctuations of the outer pore in low K+ (*Matamoros et al., 2023*; *Wang et al., 2019*), and in the model potassium channel NaK2K the outer half of the selectivity filter unravels at low K$^+$ (*Sauer et al., 2011*). On the other hand, Shaker channels are seen to conduct Na$^+$ in the absence of K$^+$ (*Melishchuk et al., 1998*). Thus, it is of interest to observe the structure of Kv1.2 under this condition.

We obtained the structure of Kv1.2$_s$ in a nominally zero K$^+$ solution, with potassium replaced with sodium in all buffers from affinity purification to size-exclusion chromatography (see Methods). We were surprised to find that it is little changed from the K$^+$-bound structure, with an essentially identical selectivity filter conformation (*Figure 4B*, *Figure 4—figure supplement 1*). Unfortunately, Na$^+$ and K$^+$ ions cannot be distinguished in the density maps, and we therefore cannot rule out the possibility that peaks in the selectivity filter region might come from contaminating K$^+$ ions. Like Kv1.2$_s$ in the usual potassium solution, the major density peaks are seen in the IS1 and IS3 ion binding sites, with somewhat lower occupancy of IS2 and IS4 (*Figure 4B and E*). Increased asymmetry in ion occupancy is expected to accompany a reduction in ion conductance (*Morais-Cabral et al., 2001*).

We conclude that the Kv1.2s selectivity filter is stable in low K$^+$ solutions, and that the channel most likely conducts Na$^+$ in the absence of potassium; both of these features have been observed in the NaK2K channel (*Sauer et al., 2011*), where the electron density profile in the selectivity filter in Na$^+$ is little changed when K$^+$ is added in a small amount—1 mM—that produces partial K$^+$ occupancy. Na$^+$ conduction is also observed in the non-inactivating KcsA E71A channel (*Mita et al., 2021*).

## Large conformational fluctuations of the W366F mutant in K$^+$-free medium

We also collected cryoEM data from the Kv1.2$_s$W17'F mutant channels, using the same procedures to replace K$^+$ with Na$^+$ during purification. The corresponding Shaker-W17'F channel stably carries large Na$^+$ and Li$^+$ currents in the absence of K$^+$ (*Starkus et al., 1998*). In 2D classification of our single-particle images we saw large variability in the structure, as if the connection between the transmembrane domain and the intracellular domains (the T1 domain and the Kvβ2 subunits) was highly flexible (*Figure 5A–B*). The intracellular domains were resolved to 3.0 Å resolution in a focused reconstruction (*Figure 5C*) but a complementary focused reconstruction of the transmembrane domain yielded only about 7 Å resolution. Nevertheless this low-resolution map matches the secondary structure of the Kv1.2$_s$ W366F mutant in K$^+$ solution (*Figure 5D*), with the possible exception of low density in the selectivity filter region (*Figure 5E–F*). We conclude that the structure of this mutant channel becomes unstable in the absence of K$^+$, as if the tight binding of K$^+$ ions is required to stabilize the altered selectivity filter.

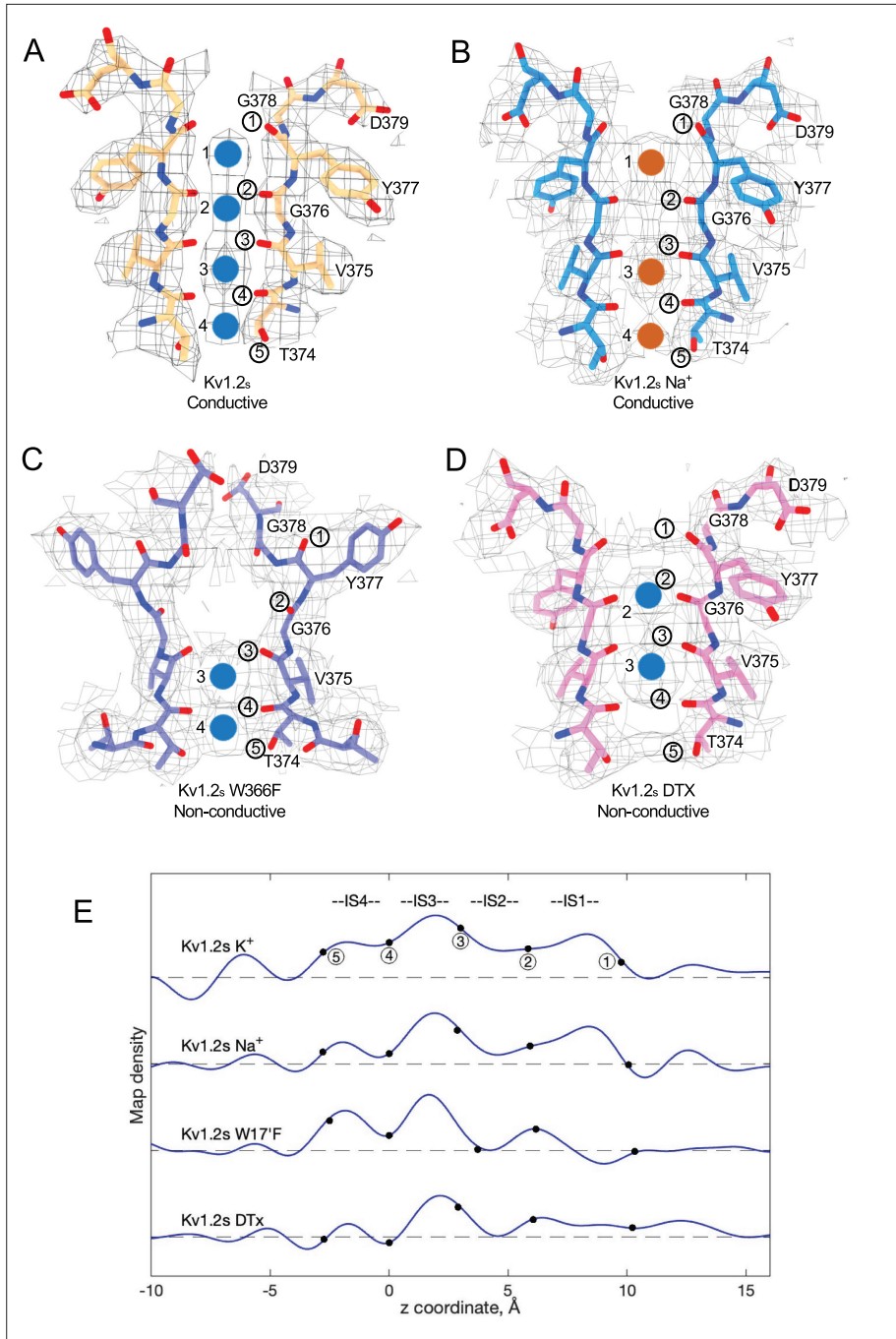

**Figure 4.** Summary of Kv1.2 conductive and non-conductive pores. Selectivity filter structures of (**A**) Kv1.2$_s$, (**B**) Kv1.2$_s$ Na$^+$-bound, (**C**) Kv1.2$_s$ W366F, and (**D**) Kv1.2$_s$-DTX. Potassium ions and sodium ions are shown as blue and orange balls, respectively. Circled numbers label the P-loop ion-coordinating oxygens. (**E**) Plots of density along the symmetry axis of the selectivity filter region. Values of the z-coordinate are relative to the position of the T25′ (T374) carbonyl oxygen. The nominal positions of the coordinating oxygens are marked and numbered as in parts A-D. Dashed baselines indicate the external solvent density. The scaling of the density traces is arbitrary. The DTx map was computed with no symmetry imposed; for the others, C4 symmetry was imposed in reconstruction.

The online version of this article includes the following figure supplement(s) for figure 4:

**Figure supplement 1.** CryoEM of Kv1.2$_s$ in Na$^+$.

**Figure supplement 2.** Comparisons of cryoEM density map and model for alpha helices in each Kv1.2 structure reported here.

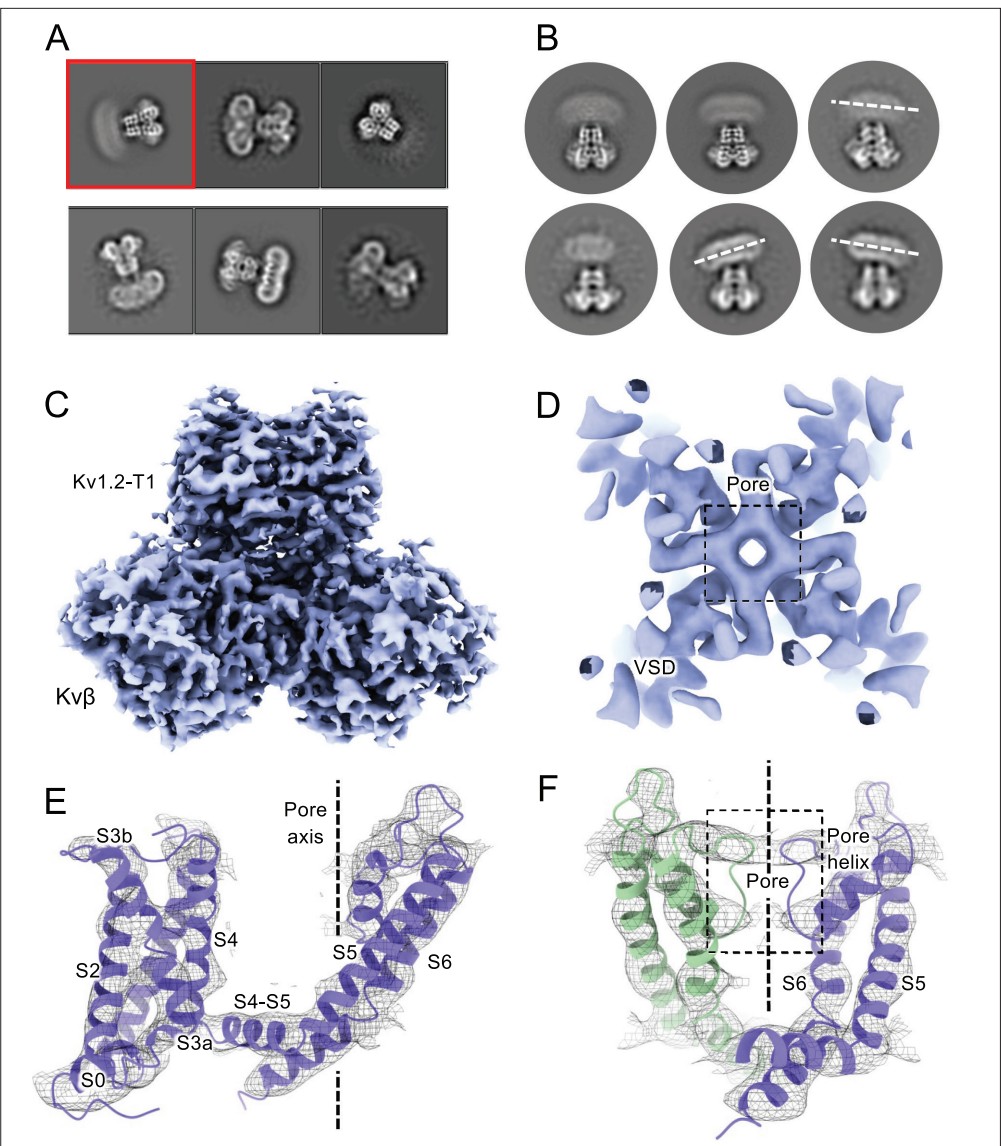

**Figure 5.** CryoEM analysis of Kv1.2 W366F in Na⁺. (**A**) Representative 2D classes. (**B**) Second round of classification of first class in (**A**), outlined in red. Wobbling of the transmembrane domain is illustrated by the white dashed lines. (**C**) Density of the intracellular domain, obtained by focused refinement. (**D**) Top view of TMD reconstruction, with resolution 7.8 Å. (**E**) Overall TMD map density fits with the Kv1.2 W366F model, shown as ribbons. (**F**) There is low density in the selectivity filter (dashed rectangle). The 2D classes and density map shown in parts A-C were obtained from a set of 239,100 particles frozen on graphene substrates. The best reconstruction of the transmembrane region (panels D-F) was however obtained from images of 68,682 particles suspended in ice, from conventional cryoEM specimens.

The online version of this article includes the following figure supplement(s) for figure 5:

**Figure supplement 1.** Representative micrograph of Kv1.2 W366F in Na⁺, demonstrating the absence of protein aggregates on the graphene substrate.

## Discussion

The potassium channel signature sequence TTVGYGD was first identified by *Heginbotham et al., 1994* as critical for potassium selectivity. The residues fold as an extended loop (the P-loop) and is part of the P-region, about 20 residues of conserved sequence located between the S5 and S6 helices. We use here a system of numbering the entire linking region between S5 and S6, as identified in Shaker by *Miller, 1991*. The P-loop (residues 24'–30' in this numbering, TTVGYGD) forms four ion binding sites

(*Doyle et al., 1998*) through which K+ ions and water molecules can pass sequentially in a knock-on fashion. The geometry and flexibility is finely tuned to allow high K+ selectivity with a high transport rate (*Noskov and Roux, 2006*; *Roux, 2005*).

In this paper we have considered the ion occupancy and conformation of the P-region under four conditions: the conducting state, a C-type inactivated state, the channel blocked by dendrotoxin, and in the absence of permeant ions. In the conducting state, the cryoEM structure of our rat Kv1.2 construct (Kv1.2s) in DDM detergent micelles agrees with the crystal structure of *Long et al., 2005* and is very similar to the structure of *Drosophila* Shaker in lipid nanodiscs (*Tan et al., 2022*). Also, apart from local differences in the chimeric region of the voltage sensor domains (VSDs), we confirm the observation (*Long et al., 2007*) that the rat Kv1.2 structure is essentially identical to the structure of the rat Kv1.2–2.1 paddle chimera as obtained from X-ray crystallography. In turn, the structure from cryoEM imaging of Kv1.2–2.1 channels in lipid nanodiscs (*Matthies et al., 2018*) is also nearly identical to the X-ray structure. Thus, the conformation of the entire transmembrane region of the Kv1.2 channel and its variants appears to be remarkably insensitive to its environment, whether lipid or detergent.

## Inactivation

After the opening of the intracellular activation gate formed by the S6 helices, C-type inactivation (*Hoshi et al., 1991*) causes channel currents to switch off spontaneously. The inactivation process is very sensitive to mutations in the selectivity filter and nearby residues of the P-region and is influenced by the state of the S6 gate (*Cuello et al., 2017*; *Li et al., 2021a*) and the VSDs (*Bassetto et al., 2021*; *Conti et al., 2016*). Low K+ concentrations, especially in the extracellular solution, accelerate inactivation (*Levy and Deutsch, 1996*).

Is C-type inactivation a reflection of selectivity filter instability? *Sauer et al., 2011* investigated the structural stability of a channel selectivity filter, using a model system based on the NaK prokaryotic

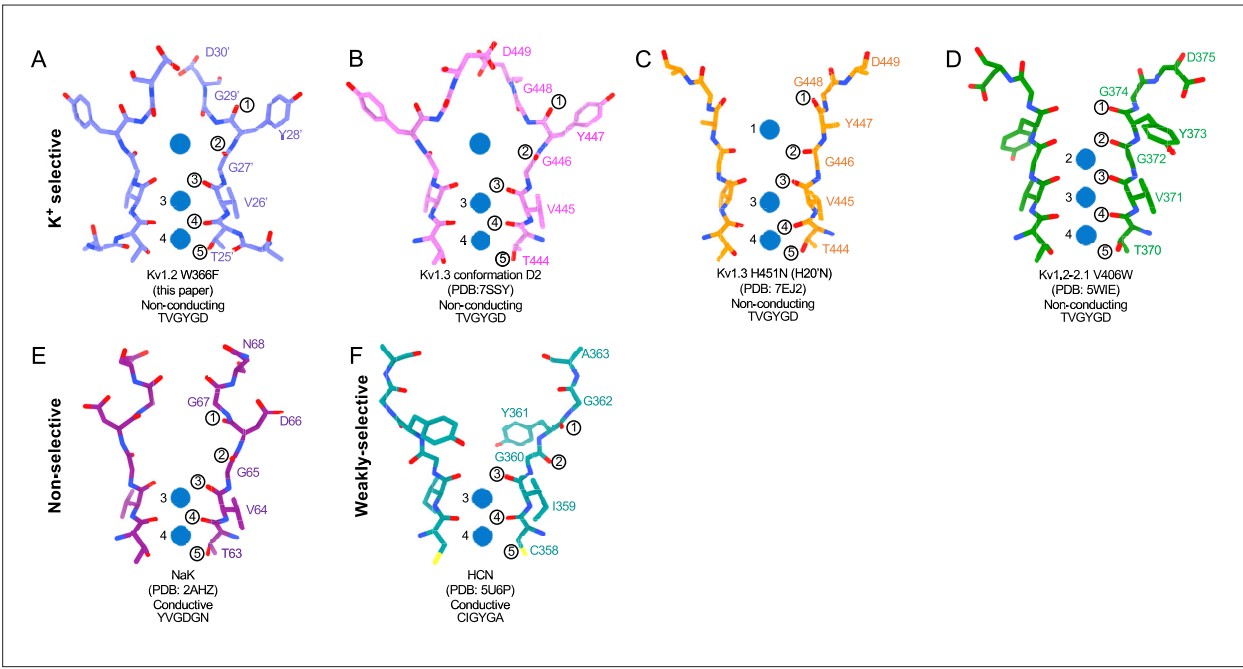

**Figure 6.** Structural comparison of Kv1.2 W366F with various channel pores. (**A-D**) Selectivity filter structures and P-loop sequences of potassium selective non-conducting channels: (**A**) Kv1.2s W366F, (**B**) Kv1.3 alternate conformation, (**C**) Kv1.3 H451N, (**D**) Kv1.2–2.1 V406W. (**E-F**) Pore regions of less-selective channels: the non-selective, conducting NaK (**E**) and the weakly-selective, conducting HCN (**F**). Potassium ions are shown as blue balls. Circled numbers enumerate the pore-forming carbonyl oxygens; carbonyl 2 faces the adjacent subunit in the clockwise direction in all but the HCN channel in panel **F**.

The online version of this article includes the following figure supplement(s) for figure 6:

**Figure supplement 1.** Structural comparison of inactivated Kv channels.

**Figure supplement 2.** Summary of conformational changes in Kv channel inactivation.

nonselective channel (*Shi et al., 2006*). NaK's selectivity filter (P-loop sequence TVGDGN) has only two ion binding sites, and opens into a wide extracellular vestibule. A similarly wide vestibule is seen in the nonselective HCN1 channel (*Lee and MacKinnon, 2017*; *Shi et al., 2006*; *Figure 6*). NaK can however be converted to a highly K$^+$ selective channel, termed NaK2K, through just the two mutations D28'Y and N30'D, yielding the P-loop sequence TVGYGD. NaK2K has the four-site selectivity filter seen in all K$^+$-selective channels, and the residues Y28' and D30' participate in a hydrogen-bonded network to stabilize the selectivity filter just as in KcsA channels (*Doyle et al., 1998*). Y28' also participates in hydrophobic packing interaction. The removal of any of these interactions results in a loss of selectivity, and unraveling of the selectivity filter, returning to the large vestibule seen in NaK channels. *Sauer et al., 2011* conclude that the standard four-ion-site, K$^+$ selective configuration of potassium channel selectivity filters is an energetically unfavorable, strained backbone conformation, and the weakening of the specific interactions that hold it in place lead to its unraveling, producing a lower-energy configuration that yields a nonselective conduction path. Similarly, the removal of K$^+$ ions from NaK2K destabilizes the selectivity filter, resulting in large fluctuations of the outer pore (*Matamoros et al., 2023*).

The H-bond interactions involving Y28' and D30' are also key to C-type inactivation in Shaker channels. Weakening those interactions produce channels that inactivate quickly or are found permanently inactivated (*Pless et al., 2013*). It is therefore no surprise that the loosening of H-bond restraints, resulting in the unraveling of the selectivity filter, yields a preferred, low-energy conformation with a large extracellular vestibule for the inactivated state of Kv channels.

In the case of Kv1.2, structural and molecular-dynamics studies have focused on the mutation W17'F (W366F in Kv1.2) which induces an inactivation-like process that is slowed in the presence of extracellular K$^+$ ions or pore blockers. In Shaker, this mutation (W434F) causes the near total loss of open-channel current (*Li et al., 2018*; *Perozo et al., 1993*; *Yang et al., 1997*); in Kv1.2, it greatly accelerates inactivation (*Suárez-Delgado et al., 2020*). A large expansion of the outer selectivity filter in W17'F mutants is seen in structures of Shaker (*Tan et al., 2022*), Kv1.2–2.1 channels (*Reddi et al., 2022*) and also in Kv1.2 as reported here. Quantified as the distance between the diagonally-positioned Y28' alpha-carbons, the expansion in Shaker is from 8.6 to 12.8 Å. In both the Kv1.2–2.1 paddle chimera (*Reddi et al., 2022*) and in Kv1.2$_s$ the expansion is approximately from 7.2 to 10.7 Å. The expanded outer pore is very similar to the large extracellular vestibules observed in NaK and HCN1 channels (*Figure 6*).

A striking feature of the inactivated state structure reported by *Reddi et al., 2022*; *Tan et al., 2022* is the high occupancy by potassium ions of the two remaining binding sites IS3 and IS4. During normal conduction, ion binding sites in the selectivity filter are usually occupied by K$^+$ and water molecules in alternation (*Morais-Cabral et al., 2001*). However, in molecular-dynamics simulations based on their Shaker W434F structure, *Tan et al., 2022* observed with high probability the simultaneous occupancy of IS3 and IS4 by two K$^+$ ions. The ions did not leave the sites during a (relatively long) 2 μs simulation with 300 mV membrane potential applied, but knock-on permeation events were observed at a low rate, two events during 1μs, at the very high potential of 450 mV. A similar result was observed in earlier MD simulations of the Kv1.2–2.1 chimera where the W17'F mutation was artificially introduced (*Conti et al., 2016*; *Kondo et al., 2018*). Those simulations did not show the large rearrangement of the outer pore, but they did demonstrate small shifts in the geometry of IS3 and IS4 that, like those seen by Tan et al., produced simultaneous occupancy by K$^+$ ions and blockage of ion permeation. We also see strong ion densities in the two remaining ion binding sites in the Kv1.2$_s$-W17'F channel (*Figures 2E and 4E*), and a similar tight binding of adjacent K$^+$ ions seems also to block K$^+$ permeation in the structure of an inactivated Kv1.3 channel (*Liu et al., 2021*).

In a study of the Shaker mutant W17'F, *Yang et al., 1997* observed, with symmetrical high K$^+$ concentrations to maximize channel activity, brief (1ms) openings of channels that occurred with a peak open probability of 10$^{-5}$ immediately following a large depolarization to +80 mV (*Yang et al., 1997*). When open, the single-channel currents were the same magnitude as the currents in wildtype channels. It is unlikely that these currents arose from rare ion-conduction events through destabilization of the tightly-bound K$^+$ ions in the inactivated configuration, as such currents would be too small to be observed. Instead, the simplest explanation is that these currents arise when structural fluctuations transiently reconstitute the normal four-site selectivity filter, and the channel conducts normally

for brief intervals. The normal selectivity filter is expected to be stabilized by bound K$^+$ ions. In the absence of external potassium ions, the peak open probability drops to about 10$^{-7}$ (*Yang et al., 1997*).

Inactivation is voltage-dependent: it occurs less in channels held at a negative membrane potential, but is enhanced in a time-dependent manner with depolarization. This was observed for example in the experiments of *Yang et al., 1997* in our recordings from Kv1.2$_s$ W17'F channels (*Figure 2—figure supplement 1*). This voltage dependence could arise from a coupling between voltage sensors and the pore domain, which would be reflected in a difference in gating-charge movement (reporting VSD conformational changes) between wildtype and W17'F channels. There is however little difference in gating-charge movement between Shaker and Shaker W17'F channels, although Shaker with certain mutations in S4 and S6 do show differences (*Bassetto et al., 2021*; *Conti et al., 2016*). A more important mechanism seems to be coupling between the S6 activation gate of the channel and the inactivation mechanism, such that steps in the physical opening of the channel would in increase the inactivation rate. Such a coupling is clearly seen in a series of KcsA X-ray structures by *Cuello et al., 2017*: with increased opening of inner bundle gate there is an increased steric clash between residues of the M2 helix and residues on the pore helix that could lead to inactivation. They identified a similar interface in the Kv1.2–2.1 structure, which provides a simple explanation for coupling between channel opening and inactivation processes.

Are the structures of the various W17'F channels valid models for C-type inactivation? In recent cryoEM studies (*Selvakumar et al., 2022*) the human Kv1.3 channel was found to spontaneously exist in one or two alternative conformations that appear to be inactivated states. The D1 conformation shows a large displacement of D30' and is reminiscent of the cryoEM structure of the Kv1.3 rapidly-inactivating mutant H20'N (*Liu et al., 2021*). It exhibits a partially twisted P-loop (*Figure 6*, *Figure 6—figure supplement 2B*) as if it were an intermediate state between open and the W17'F structures. Meanwhile the D2 conformation is very similar to that of the W17'F variants considered here. This strongly supports the widely-held view that these mutant channels are in a conformation close to, if not identical to, a true C-type inactivated state.

Why is inactivation much more complete in the Shaker mutant W17'F than in Kv1.2 or Kv1.2–2.1? Clearly one can compare the stabilities of the conducting and the inactivated conformations. *Wu et al., 2022* mutated several of the residues that form important interactions in these channels, and found that the inactivation behaviors of Kv1.2 and Kv1.2–2.1 are different from that of Shaker. In the conducting state, the H-bond networks stabilizing the key residues Y28' and D30' in Shaker and Kv1.2 very similar, with the only obvious difference being at position 22' where Shaker has Thr but Kv1.2 has Ser. It turns out that the mutation S22'T in the Kv1.2 background has little effect on inactivation; thus to a first approximation the conducting state is equivalently stabilized.

In the inactivated state structures, all obtained with the W17'F mutation, a variety of H-bond interactions arise in the linker sequence between the P-loop and the S6 helix, with the variety arising from different conformations of Loop 1 and Loop 2, two regions of the linker (*Figure 6—figure supplement 1*). Details of the interactions are discussed in the Figure legend. We conclude that a quantitative evaluation of the stability of the inactivated state will require more than simply an enumeration of H-bonds in this region of the protein sequence.

## Block of the channel by dendrotoxin

Potassium currents can be inhibited by five distinct families of toxins. The spider toxins Hanatoxin, SGTx1 and VSTx1 interact with the voltage sensors of Kv channels to inhibit voltage-dependent channel activation (*Lee and MacKinnon, 2004*; *Wang et al., 2004*). Other toxins bind directly to the extracellular mouth of the channel to block current. The cone-snail toxin Cs1 binds to the extracellular surface of Kv channels and is thought to block currents by inducing a conformational change in the protein, collapsing the selectivity filter (*Karbat et al., 2019*). In X-ray crystal structures the scorpion toxin Charybdotoxin is seen to block Kv1.2-paddle chimera channels by inserting a lysine residue directly into the selectivity filter, disrupting the ion-binding site IS1 to inhibit ion flow (*Banerjee et al., 2013*). The same mechanism holds for block of Kv1.3 by the sea anemone toxin ShK as studied by cryoEM (*Selvakumar et al., 2022*); in the latter two cases, the lysine terminal amine interacts with the Y17' carbonyls which form the top of binding site IS1. We now report that Kv1.2 block by the snake toxin α-Dendrotoxin (α-DTx) is by a similar mechanism.

Obtaining the structure of an asymmetric ligand bound to a four-fold symmetric channel is challenging, and the four structural studies that have been done have used different strategies. In their X-ray crystallographic study, *Banerjee et al., 2013* were able to interpret and model at low resolution the fourfold superimposed toxin density, using heavy-atom derivatives in a series of crystal structures to establish the pose of the toxin molecule. The difficulty of modeling the CTx molecule precluded assigning a precise location to the Lys22 channel-blocking sidechain; nevertheless, in the 2.5 Å map of the selectivity filter are seen three well-resolved, discrete ion densities in the sites IS2, IS3 and IS4, and the authors conclude that the toxin interaction prevents ion occupancy of the IS1 site. A cryoEM study that yielded a similar C4 symmetric reconstruction of Kv1.3 with bound dalazatide, a derivative of the ShK toxin, was obtained by *Tyagi et al., 2022*. The C4-symmetric reconstruction did not yield an interpretable map of the toxin, but clear ion densities in IS2, IS3, and IS4 were visible.

*Selvakumar et al., 2022* also obtained a structure of ShK bound to Kv1.3, but in this case the ShK toxin was fused to a Fab fragment; this yielded an asymmetric ligand large enough to allow C1 reconstruction from the cryoEM images. The resulting 3.4 Å map allowed the toxin K22 sidechain to be modeled entering the pore and extending to near the Y17' carbonyl oxygens.

From our cryoEM data we obtained a C1 map of α-DTx bound to Kv1.2$_s$ at 3.2 Å; this was done by extensive image processing and was helped by the larger size of α-DTx (59 residues as compared to 35 for ShK). With this map, the side chain of the toxin residue K5 could be traced, apparently with the terminal amine directly coordinated by the Y17' carbonyls. A surprise in this map are the positions of the K$^+$ ion peaks in the selectivity filter (*Figure 4E*). Ion density is visible in IS3, and somewhat lower density is seen in IS1; there is no density peak in IS2. The different pattern of ion densities with DTx compared to CTx, having two rather than three peaks in the selectivity filter, implies that the interaction between the lysine side chain and the selectivity filter is in some way different in DTx compared to the other toxins.

## The selectivity filter in the absence of K$^+$

In the absence of K$^+$, large voltage-dependent Na$^+$ currents are observed in wildtype Kv2.1 channels (*Korn and Ikeda, 1995*) and in Shaker channels (*Melishchuk et al., 1998*).

The Kv1.2 structure reported here with Na$^+$ replacing essentially all K$^+$ in the solution shows a selectivity filter conformation that is little changed from the K$^+$ structure (*Figure 5B*). Ion densities are seen in the ion-binding sites, and the occupancies of the sites are altered, but the limited resolution precludes an analysis of the coordination of ions.

The Shaker-W17'F channel in the absence of K$^+$ stably carries large Na$^+$ and Li$^+$ currents (*Starkus et al., 1997*). These currents are blocked by low concentrations of K$^+$, and this would be expected as potassium binding to the sites IS3 and IS4 in this mutant is very tight (*Tan et al., 2022*). We therefore sought to obtain the structure of the corresponding Kv1.2-W17'F channel in Na$^+$ solution (*Figure 5*). In 2D classes, the complex appeared to be highly unstable, with large variations in the 'wobble' angle between the transmembrane and cytoplasmic domains. Attempts to recover the structure of the transmembrane region by focused refinement yielded only low-resolution structural information. We conclude that this channel, at least as solubilized in detergent, has a highly unstable conformation when K$^+$ ions are absent.

## Methods
### Kv1.2-Kvβ expression and purification

The rat Kv1.2 alpha-subunit constructs were derived from that of *Long et al., 2005*, having the full-length (GenBank: X16003) sequence containing the mutation N207Q to eliminate a glycosylation site. Our Kv1.2$_s$ construct contained the additional mutations L15H and G198S in unstructured regions; the inactivating construct contained the further mutation W366F. To each of these was added at the N-terminus two tandem Strep tags (sequence WSHPQFEK) separated by a Gly-Ser linker. The beta-subunit construct was derived from the rat beta2 core, residues 36–357 (*Gulbis et al., 1999*). To eliminate very strong electrostatic interactions between the many Lys residues on the 'bottom' of the beta-subunits and the negatively-charged carbon film or graphene substrates, we mutated to glutamine the five lysine residues at positions 94, 104–106 and 258 of the beta2 subunit. In our structures, the fold of

the beta subunits containing these mutations is indistinguishable from that of the wildtype structure of *Long et al., 2007*.

To express both subunits in a single construct, Kv1.2 and Kvβ genes were separately inserted into pPicZ-B (Thermo Fisher) vectors between the XhoI and AgeI sites in the multi-restriction-sites region. Subsequently the pPicZ-B vector carrying Kvβ was opened at BglII and BamHI restriction sites, and the insert including the AOX1 promoter was inserted into the other pPicZ-B vector, upstream of the AOX1-driven Kv1.2 cDNA, at the BglII site.

The *P. pastoris* strain SMD1168 (Thermo Fisher Scientific) was electroporated with the pPicZ-B plasmids. YPDS (yeast extract, peptone, dextrose, and sorbitol) plates containing Zeocin (800 μg/ml) were used to select transformants. Stocks were stored in 15% glycerol at –80 °C.

The expression and purification procedures were based on those of *Long et al., 2005*. Briefly, we inoculated 1 liter of BMGY medium (this and the other yeast media were obtained from Invitrogen) with 10 ml of an overnight culture grown in YPD medium with Zeocin (100 μg/ml). After centrifugation (5500 × *g*, 10 min), the cells were transferred to BMMY medium including 0.5% (v/v) methanol and grown for 24 hr. After adding an additional 0.5% methanol to increase protein expression, cells were grown for 2–3 days. Cells were centrifuged (5500 × *g*, 20 min), the pellet was frozen and stored at –80 °C.

Thawed cells were lysed at 4° in a French Press. One gram of cell lysate was resuspended in 5 mL of Lysis buffer consists of 100 mM Tris-HCl (pH 8.0), 150 mM KCl, 300 mM sucrose, 5 mM EDTA, 0.05 mg deoxyribonuclease I, 1 mM $MgCl_2$, 1 mM phenylmethylsulfonyl fluoride, 1 mg/ml each of leupeptin, pepstatin and aprotin, and 0.1 mg/mL of soy trypsin inhibitor. Lysis buffer plus 30 mM dodecylmaltoside (DDM) was used to solubilize the membranes for 3 hr at room temperature. Centrifugation (13,000× *g*, 20 min) separated the unsolubilized material. The supernatant was added to Strep-Tactin Sepharose beads (IBA Lifesciences) preequilibrated with Membrane Resuspension Buffer (100 mM Tris-HCl pH 8.0, 150 mM KCl, 3 mM TCEP, 5 mM EDTA, 10 mM beta-mercaptoethanol, and 5 mM DDM). The bead slurry (approximately 1 ml) was incubated 1 hr at 4 °C with gentle rotation. A column was used to collect the beads after incubation, followed by washing with 14 volumes of Wash Buffer, which was Membrane Resuspension Buffer with added lipids (0.1 mg/ml of the mixture 3:1:1 of POPC:POPE:POPG). The addition of 10 mM desthiobiotin was used to elute the bound protein. The eluted protein was concentrated with a Millipore Amicon Ultra 100 K filter and further purified by size-exclusion chromatography (SEC) on a Superose S6 column pre-equilibrated with Running Buffer: 20 mM tris-HCl (pH 7.5), 150 mM KCl, 2 mM TCEP, 10 mM DTT, 1 mM EDTA, 1 mM PMSF, 1 mM dodecylmaltoside (Anatrace, anagrade) and 0.1 mg/ml lipid mixture (3:1:1 of POPC:POPE:POPG). We pooled fractions containing both alpha and beta subunits, and when necessary concentrated the protein to 1~2 mg/ml (Amicon Ultra 100 KDa, Millipore). For experiments under $K^+$-free conditions, the protein bound to Strep-Tactin was washed with 5 column volumes of $K^+$-free Wash Buffer (150 mM NaCl replacing KCl) and eluted in 3 mL of $K^+$-free elution buffer. As for SEC, the column was first washed with three volumes of $K^+$-free Running Buffer before running the protein sample.

Protein expression was monitored by western blot using an anti-strep-tag antibody, Abnova PAB16603.

## Oocyte expression and voltage-clamp recordings

The $Kv1.2_s$ alpha subunit or its W366F mutant construct was cloned into a pcDNA3.1 vector. RNA was prepared from the XbaI-linearized plasmid using T7 RNA polymerase. Oocytes were obtained from a *Xenopus laevis* frog (Nasco Education) and were defolliculated by collagenase treatment, injected with cRNA and stored in ND96 solution (96 mM NaCl, 2 mM KCl, 1.8 mM CaCl2, 1 mM $MgCl_2$, 5 mM HEPES, pH 7.4 with NaOH) at 18 °C. Recordings were done at room temperature, 5–6 days post-injection in ND96 solution or KD96, the same solution but made with 96 mM KCl, 2 mM NaCl and KOH. Two-electrode voltage clamp recordings employed an OC-725C amplifier (Warner Instruments).

## CryoEM specimen preparation, data acquisition, and processing

Quantifoil holey carbon grids (R1.2/1.3, 300 mesh, Au) were glow-discharged at 15 mA for 1 min with carbon side facing upwards in the chamber. The chamber of the freezing apparatus (Vitrobot Mark IV, Thermo Fisher Scientific) was preequilibrated at 16 °C and 100% humidity. A 3 μL droplet of protein sample was applied to the carbon side of each grid and blotted for 3 or 5 s with zero blotting force

after 15 s wait time before plunge-freezing in liquid ethane. For the DTX bound sample, about 400 nM α-DTX (Alomone Labs) was mixed with Kv1.2$_s$ protein 30 min prior to cryoEM grid preparation.

For the Kv1.2$_s$-Na$^+$, W366F-Na$^+$, and DTX-bound samples, graphene-covered grids were used to allow lower protein concentrations (0.05–0.1 mg/ml) to be used. These were prepared from commercial graphene (Trivial Transfer Graphene, single layer; ACS Material LLC) using the recommended transfer protocol and floated onto quantifoil grids. To make the graphene layer hydrophilic, grids were then plasma-cleaned (Gatan 950 Advanced Plasma System) under Ar/O$_2$ at 10 W for 10 s with the graphene coated surface facing upwards in the chamber (*Fan et al., 2019*). Vitrification was conducted as described above but with 30 s wait time.

A table of cryoEM data collection, refinement and validation statistics is given in *Supplementary file 1*. CryoEM samples were imaged on a Titan Krios microscope (Thermo Fisher Scientific) operated at 300 keV, equipped with a Bio Quantum energy filter (20 eV slit width) and K3 camera (Gatan). Micrographs were collected using super-resolution mode, physical pixel size 1.068 Å, with SerialEM software (*Schorb et al., 2019*) using image shift patterns and one shot per hole. A total of 6335, 8310, 4507, 2573, and 9628 micrographs were collected for datasets of WT, W366F, DTX-bound, Na$^+$-bound WT, and the Na$^+$ bound W366F transmembrane region.

Processing for each dataset is summarized in the Figure Supplements to *Figures 1–4*. Beam-induced motion was corrected by MotionCor2 (*Zheng et al., 2017*) and CTF values were estimated by Gctf (*Zhang, 2016*). Particles were picked either by RELION reference-free Laplacian-of-Gaussian autopicking; an adversarial-template-based program DemoPicker written by F.J.S.; or RELION reference-based auto-picking. All further processing used RELION 3.1. Picked particles were extracted and subjected to rounds of 2D and 3D classification before refinement. Transmembrane-domain masks were used for focused refinement to optimize the resolution of the transmembrane region. Beam-tilt and per-particle CTF refinement followed by per-particle motion correction were applied to further improve the resolution.

Analysis of the DTx-Kv1.2$_s$ complex started with a C4 reconstruction of the complex from 323 k particles. We then performed a rough docking into the Kv1.2$_s$ apo density of a toxin map created from the 1DTX model. The toxin pose was chosen not to be inconsistent with the symmetrized toxin density visible in the DTx-Kv1.2$_s$ C4 map. Four of these composite maps were created, with the toxin pose rotated 90° each time about the z-axis. These were filtered to 20 Å resolution for use as 3D references. Meanwhile a C4 symmetry expansion of the 323 k particles was performed, resulting in a 1.294 M particle stack. A soft cylindrical mask of radius 24 Å and height 48 Å was created, centered on the z-axis at the interface between the toxin and the pore region. Using this mask and the four 3D references, we performed 3D classification without alignment in RELION, and chose one of the four resulting classes as a C1 reconstruction of the masked region. Using the corresponding set of 312 k particles, we performed C1 homogeneous reconstruction in CryoSPARC to yield the refined structure at 3.2 Å resolution.

## Protein model building, refinement, and structural analysis

An Alpha-Fold predicted rat Kv1.2 model (alphafold.ebi.ac.uk/entry/Q09081) was used to build the Kv1.2 native atomic model. The W366F, DTX-bound, and Na$^+$-bound models were subsequently built from this Kv1.2 model. Model fitting was performed with the CCPEM program suite. Initial docking was performed with Molrep; obvious outliers of were manually fixed in Coot 0.9 (*Emsley et al., 2010*) and real space refinement used Refmac 5 (*Murshudov et al., 2011*) and Phenix (*Liebschner et al., 2019*). Density map rendering, analysis and figure preparation were done with UCSF Chimera and ChimeraX (*Goddard et al., 2018*; *Pettersen et al., 2004*). A summary of data refinement and validation statistics is provided in *Supplementary file 1*.

## Acknowledgements

We thank Marc Llaguno and Jianfeng Lin for aid with microscopy for screening on the Glacios microscopes, and Shenping Wu for aid with the Titan Krios data collection. We are grateful to Yufeng Zhou (University of Pennsylvania) and David Fedida (University of British Columbia) for providing the original constructs and consultation on expression constructs. We are grateful to the Yale Center for Research Computing and to the SBGrid Consortium (*Herre et al., 2024*) for software support.

## Additional information

### Funding

| Funder | Grant reference number | Author |
|---|---|---|
| National Institutes of Health | R01NS021501 | Fred J Sigworth |
| National Institutes of Health | S10OD023603 | Fred J Sigworth |

The funders had no role in study design, data collection and interpretation, or the decision to submit the work for publication.

### Author contributions

Yangyu Wu, Conceptualization, Data curation, Formal analysis, Validation, Investigation, Visualization, Methodology, Writing - original draft, Writing – review and editing; Yangyang Yan, Resources, Investigation; Youshan Yang, Investigation, Visualization, Methodology; Shumin Bian, Alberto Rivetta, Ken Allen, Investigation, Methodology; Fred J Sigworth, Formal analysis, Funding acquisition, Validation, Methodology, Project administration, Writing – review and editing

### Author ORCIDs

Yangyu Wu (ID) https://orcid.org/0000-0001-8064-6132
Fred J Sigworth (ID) https://orcid.org/0000-0002-7178-8494

Reviewer #1 (Public Review): https://doi.org/10.7554/eLife.89459.4.sa1
Reviewer #2 (Public Review): https://doi.org/10.7554/eLife.89459.4.sa2
Reviewer #3 (Public Review): https://doi.org/10.7554/eLife.89459.4.sa3
Author response https://doi.org/10.7554/eLife.89459.4.sa4

## Additional files

### Supplementary files

Supplementary file 1. CryoEM data collection, refinement and validation statistics.

MDAR checklist

### Data availability

DNA expression vectors and P. pastoris clones are available from the authors upon request. The Matlab code for the particle picking program DemoPIcker is publicly available at https://github.com/fsigworth/aEMCodeRepository/ (copy archived at *Sigworth, 2025*) in the subdirectory Teaching/PartPickingDemo.

The following datasets were generated:

| Author(s) | Year | Dataset title | Dataset URL | Database and Identifier |
|---|---|---|---|---|
| Wu Y, Sigworth FJ | 2023 | Voltage gated potassium ion channel Kv1.2 in potassium | https://www.rcsb.org/structure/8VC6 | RCSB Protein Data Bank, 8VC6 |
| Wu Y, Sigworth FJ | 2023 | Voltage gated potassium ion channel Kv1.2 in potassium | https://www.ebi.ac.uk/emdb/EMD-43134 | Electron Microscopy Data Bank, EMD-43134 |
| Wu Y, Sigworth FJ | 2023 | Voltage gated potassium ion channel Kv1.2 W366F, C-type inactivated | https://www.rcsb.org/structure/8VCH | RCSB Protein Data Bank, 8VCH |

*Continued on next page*

*Continued*

| Author(s) | Year | Dataset title | Dataset URL | Database and Identifier |
|---|---|---|---|---|
| Wu Y, Sigworth FJ | 2023 | Voltage gated potassium ion channel Kv1.2 W366F, C-type inactivated | https://www.ebi.ac.uk/emdb/EMD-43136 | Electron Microscopy Data Bank, EMD-43136 |
| Wu Y, Sigworth FJ | 2023 | Voltage gated potassium ion channel Kv1.2 in complex with DTx | https://www.rcsb.org/structure/8VC3 | RCSB Protein Data Bank, 8VC3 |
| Wu Y, Sigworth FJ | 2023 | Voltage gated potassium ion channel Kv1.2 in complex with DTx | https://www.ebi.ac.uk/emdb/EMD-43131 | Electron Microscopy Data Bank, EMD-43131 |
| Wu Y, Sigworth FJ | 2023 | Voltage gated potassium ion channel Kv1.2 in sodium | https://www.rcsb.org/structure/8VC4 | RCSB Protein Data Bank, 8VC4 |
| Wu Y, Sigworth FJ | 2023 | Voltage gated potassium ion channel Kv1.2 in sodium | https://www.ebi.ac.uk/emdb/EMD-43133 | Electron Microscopy Data Bank, EMD-43133 |

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
