## [Editor Report · eLife assessment]

This **important** manuscript presents several structures of the Kv1.2 voltage-gated potassium channel, based on state-of-the-art cryoEM techniques and algorithms. The authors present **solid** evidence for structures of an inactivating mutant of Kv1.2, DTX-bound Kv1.2 and of Kv1.2 in potassium-free solution (with presumably sodium ions bound within the selectivity filter). These structures advance our knowledge of the molecular basis of the slow inactivation process of potassium channels.

---

## [Referee Report · Reviewer #1 (Public Review)]

In this manuscript by Wu et al., the authors present the high resolution cryoEM structures of the WT Kv1.2 voltage-gated potassium channel. Along with this structure, the authors have solved several structures of mutants or experimental conditions relevant to the slow inactivation process that these channels undergo and which is not yet completely understood.

One of the main findings is the determination of the structure of a mutant (W366F) that is thought to correspond to the slow inactivated state. These experiments confirm results in similar mutants in different channels from Kv1.2 that indicate that inactivation is associated with an enlarged selectivity filter.

Another interesting structure is the complex of Kv1.2 with the pore blocking toxin Dendrotoxin 1. The results shown in the revised version indicate that the mechanism of block is similar to that of related blocking-toxins, in which a lysine residue penetrates in the pore. Surprisingly, in these new structures, the bound toxin results in a pore with empty external potassium binding sites.

The quality of the structural data presented in this revised manuscript is very high and allows for unambiguous assignment of side chains. The conclusions are supported by the data. This is an important contribution that should further our understanding of voltage-dependent potassium channel gating. In the revised version, the authors have addressed my previous specific comments.

---

## [Referee Report · Reviewer #2 (Public Review)]

Cryo_EM structures of the Kv1.2 channel in the open, inactivated, toxin complex and in Na+ are reported. The structures of the open and inactivated channels are merely confirmatory of previous reports. The structures of the dendrotoxin bound Kv1.2 and the channel in Na+ are new findings that will of interest to the general channel community.

Review of the resubmission:

I thank the authors for making the changes in their manuscript as suggested in the previous review. The changes in the figures and the additions to the text do improve the manuscript. The new findings from a further analysis of the toxin channel complex are welcome information on the mode of the binding of dendrotoxin.

---

## [Referee Report · Reviewer #3 (Public Review)]

Wu et al. present cryo-EM structures of the potassium channel Kv1.2 in open, C-type inactivated, toxin-blocked and presumably sodium-bound states at 3.2 Å, 2.5 Å, 2.8 Å, and 2.9 Å. The work builds on a large body of structural work on Kv1.2 and related voltage-gated potassium channels. The manuscript presents a plethora of structural work, and the authors are commended on the breadth of the studies. The structural studies are well-executed. Although the findings are mostly confirmatory, they do add to the body of work on this and related channels. Notably, the authors present structures of DTx-bound Kv1.2 and of Kv1.2 in a low concentration of potassium (which may contain sodium ions bound within the selectivity filter). These two structures add considerable new information. The DTx structure has been markedly improved in the revised version and the authors arrive at well-founded conclusions regarding its mechanism of block. Overall, the manuscript is well-written, a nice addition to the field, and a crowning achievement for the Sigworth lab.

---

## [Author Response]

The following is the authors’ response to the previous reviews.

**Public Reviews:**

**Reviewer #1 (Public Review):**
In this manuscript by Wu et al., the authors present the high resolution cryoEM structures of the WT Kv1.2 voltagegated potassium channel. Along with this structure the authors have solved several structures of mutants or experimental conditions relevant to the slow inactivation process that these channels undergo and which is not yet completely understood.One of the main findings is the determination of the structure of a mutant (W366F) that is thought to correspond to the slow inactivated state. These experiments confirm results in similar mutants in different channels from Kv1.2 that indicate that inactivation is associated with an enlarged selectivity filter.Another interesting structure is the complex of Kv1.2 with the pore blocking toxin Dendrotoxin 1. The results shown in the revised version indicate that the mechanism of block is similar to that of related blocking-toxins, in which a lysine residue penetrates in the pore. Surprisingly, in these new structures, the bound toxin results in a pore with empty external potassium binding sites.The quality of the structural data presented in this revised manuscript is very high and allows for unambiguous assignment of side chains. The conclusions are supported by the data. This is an important contribution that should further our understanding of voltage-dependent potassium channel gating. In the revised version, the authors have addressed my previous specific comments, which are appended below.(1) In the main text's reference to Figure 2d residues W18' and S22' are mentioned but are not labeled in the insets.

This has been fixed: line 229, p. 9.

(2) On page 8 there is a discussion of how the two remaining K+ ions in binding sites S3 and S4 prevent permeation K+ in molecular dynamics. However, in Shaker, inactivated W434F channels can sporadically allow K+ permeation with normal single-channel conductance but very reduced open times and open probability at not very high voltages.

This is noted in the discussion Lines 497-500, p. 18

(3) The structures of WT in the absence of K+ shows a narrower selectivity filter, however Figure 4 does not convey this finding. In fact, the structure in Figure 4B is constructed in such an angle that it looks as if the carbonyl distances are increased, perhaps this should be fixed. Also, it is not clear how the distances between carbonyls given in the text on page 12 are measured. Is it between adjacent or kitty-corner subunits?

We have changed Fig. 4B to show the same view as in Fig. 4A. In the legend we explain that opposing subunits are shown. We no longer give distances, in view of the lack of detectable carbonyl densities.

(4) It would be really interesting to know the authors opinion on the driving forces behind slow inactivation. For example, potassium flux seems to be necessary for channels to inactivate, which might indicate a local conformational change is the trigger for the main twisting events proposed here.

We address this in the Discussion, line 506-523, pp. 18-19.

**Reviewer #2 (Public Review)**:Cryo_EM structures of the Kv1.2 channel in the open, inactivated, toxin complex and in Na+ are reported. The structures of the open and inactivated channels are merely confirmatory of previous reports. The structures of the dendrotoxin bound Kv1.2 and the channel in Na+ are new findings that will of interest to the general channel community.Review of the resubmission:I thank the authors for making the changes in their manuscript as suggested in the previous review. The changes in the figures and the additions to the text do improve the manuscript. The new findings from a further analysis of the toxin channel complex are welcome information on the mode of the binding of dendrotoxin.A few minor concerns:(1) Line 93-96, 352: I am not sure as to what is it the authors are referring to when they say NaK2P. It is either NaK or NaK2K. I don't think that it has been shown in the reference suggested that either of these channels change conformation based on the K+ concentration. Please check if there is a mistake and that the Nichols et. al. reference is what is being referred to.

Thank you for noticing the error. We meant NaK2K and we have changed this throughout.

(2) Line 365: In the study by Cabral et. al., Rb+ ions were observed by crystallography in the S1, S3 and S4 site, not the S2 site. Please correct.

Thank you. We have re-written this section, lines 364-381, pp. 13-14.

**Reviewer #3 (Public Review):**
Wu et al. present cryo-EM structures of the potassium channel Kv1.2 in open, C-type inactivated, toxin-blocked and presumably sodium-bound states at 3.2 Å, 2.5 Å, 2.8 Å, and 2.9 Å. The work builds on a large body of structural work on Kv1.2 and related voltage-gated potassium channels. The manuscript presents a plethora of structural work, and the authors are commended on the breadth of the studies. The structural studies are well-executed. Although the findings are mostly confirmatory, they do add to the body of work on this and related channels. Notably, the authors present structures of DTx-bound Kv1.2 and of Kv1.2 in a low concentration of potassium (which may contain sodium ions bound within the selectivity filter). These two structures add considerable new information. The DTx structure has been markedly improved in the revised version and the authors arrive at well-founded conclusions regarding its mechanism of block. Regarding the Na+ structure, the authors claim that the structure with sodium has "zero" potassium - I caution them to make this claim. It is likely that some K+ persists in their sample and that some of the density in the "zero potassium" structure may be due to K+ rather than Na+. This can be clarified by revisions to the text and discussion. I do not think that any additional experiments are needed. Overall, the manuscript is well-written, a nice addition to the field, and a crowning achievement for the Sigworth lab.Most of this reviewer's initial comments have been addressed in the revised manuscript. Some comments remain that could be addressed by revisions of the text.Specific comments on the revised version:Quotations indicate text in the manuscript.(1) "While the VSD helices in Kv1.2s and the inactivated Kv1.2s-W17'F superimpose very well at the top (including the S4-S5 interface described above), there is a general twist of the helix bundle that yields an overall rotation of about 3o at the bottom of the VSD."Comment: This seemed a bit confusing. I assume the authors aligned the complete structures - the differences they indicate seem to be slight VSD repositioning relative to the pore rather than differences between the VSD conformations themselves. The authors may wish to clarify. As they point out in the subsequent paragraph, the VSDs are known to be loosely associated with the pore.

We aligned the VSDs alone, and it is a twist of the VSD helix bundle.

This is now clarified in lines 269-273, p. 10.

(2) Comment: The modeling of DTx into the density is a major improvement in the revision. Figure 3 displays some interactions between the toxin and Kv1.2 - additional side views of the toxin and the channel might allow the reader to appreciate the interactions more fully. The overall fit of the toxin structure into the density is somewhat difficult to assess from the figure. (The authors might consider using ChimeraX to display density and model in this figure.)

We have added new panels, and stereo pairs, to Figure 3.

(3) "We obtained the structure of Kv1.2s in a zero K+ solution, with all potassium replaced with sodium, and were surprised to find that it is little changed from the K+ bound structure, with an essentially identical selectivity filter conformation (Figure 4B and Figure 4-figure supplement 1)."Comment: It should be noted in the manuscript that K+ and Na+ ions cannot be distinguished by the cryo-EM studies - the densities are indistinguishable. The authors are inferring that the observed density corresponds to Na+ because the protein was exchanged from K+ into Na+ on a gel filtration (SEC) column. It is likely that a small amount of K+ remains in the protein sample following SEC. I caution the authors to claim that there is zero K+ in solution without measuring the K+ content of the protein sample. Additionally, it should be considered that K+ may be present in the blotting paper used for cryo-EM grid preparation (our laboratory has noted, for example, a substantial amount of Ca2+ in blotting paper). The affinity of Kv1.2 for K+ has not been determined, to my knowledge - the authors note in the Discussion that the Shaker channel has "tight" binding for K+. It seems possible that some portion of the density in the selectivity filter could be due to residual K+. This caveat should be clearly stated in the main text and discussion. More extensive exchange into Na+, such as performing the entire protein purification in NaCl, or by dialysis (as performed for obtaining the structure of KcsA in low K+ by Y. Zhou et al. & Mackinnon 2001), would provide more convincing removal of K+, but I suspect that the Kv1.2 protein would not have sufficient biochemical stability without K+ to endure this treatment. One might argue that reduced biochemical stability in NaCl could be an indication that there was a meaningful amount of K+ in the final sample used for cryo-EM (or in the particles that were selected to yield the final high-resolution structure).

We now explain in the Methods section, in more detail the steps taken to avoid any residual Na+ contamination during purification, lines 683-687, pp. 24-25. We have changed the text to point out that the ion species cannot be distinguished in the maps, and note results in NaK2K and KcsA (lines 368-381, pp. 13-14).

We note that the same procedures to remove K+ were used for the Kv1.2sW17’F structure (line 385, p. 14). We qualify the ion replacement to say that Na+ replaces “essentially” all K+ (line 607, p. 21).

(4) Referring to the structure obtained in NaCl: "The ion occupancy is also similar, and we presume that Kv1.2 is a conducting channel in sodium solution."Comment: Stating that "Kv1.2 is a conducting channel in sodium solution" and implying that conduction of Na+ is achieved by an analogous distribution of ion binding sites as observed for K+ are strong statements to make - and not justified by the experiments provided. Electrophysiology would be required to demonstrate that the channel conducts sodium in the absence of K+. More complete ionic exchange, better control of the ionic conditions (Na+ vs K+), and affinity measurements for K+ would be needed to determine the distribution of Na+ in the filter (as mentioned above). At minimum, the authors should revise and clarify what the intended meaning of the statement "we presume that Kv1.2 is a conducting channel in sodium solution". As mentioned above, it seems possible/likely that a portion of the density in the filter may be due to K+.

We now present a more detailed argument (lines 376 to 381, pp. 13-14.)

**Recommendations for the authors:**

**Reviewing Editor:**
After consultation, the reviewers agree that, although the authors have answered most of the comments raised in the previous review, there remains a concern about the structure obtained in the presence if Na. Given that Kv1.2 is more reluctant to slow inactivation, the conducting structure in Na+ could be due to this fact or that it really has higher affinity for K+ than Na+. In the presence of even a small contamination by K+, this ion could thus occupy the selectivity filter, resulting in an open conformation. The authors should clearly state the steps taken to ensure no contamination by K+. It is also possible that indeed the open structure occurs even in the presence of Na+ in the selectivity filter. This should be also discussed, given that this has been observed in other potassium channel structures.
**Reviewer #1 (Recommendations For The Authors):**
In this revised version of the manuscript, the authors have adequately addressed my previous points and improved the clarity and readability of the manuscript. This is a compelling work that shows inactivated structures if the Kv1.2 potassium channel, especially interesting is a structure in the absence of extracellular potassium ions, that can help understand how a reduction in the availability of these ions speed up entrance into the inactivated state in these ion channels.I would just recommend that in the absence of functional data (current recordings) when potassium is removed, the authors just use caution in ascribing this structure to an inactivated state. Also, it should be mentioned that the observed ion densities observed in the pore cannot unambiguously be identified as sodium ions.
**Reviewer #3 (Recommendations For The Authors):**
(1) "The nearby Leu9 is also important as its substitution by alanine also decreases affinity 1000-fold, but we observe no contacts between this residue and residues of the Kv1.2s channel."Comment: It seems early in the text to mention the potential interaction of Leu9 to the channel structure. The authors may wish to discuss Leu9 later in the manuscript - a figure showing the location of Leu9 would strengthen the statement. Any hypothesis on why mutation of it has such a profound effect?Add a figure panel showing Leu9 position.

We have rewritten the text as suggested, and have identified Leu9 in several panels of Fig. 3.

(2) "The X-ray structure of a-DTX (Figure 3A)"Comment: The authors may wish to cite a reference to this X-ray structure.

We now cite Skarzynski (1992) on line 321, p. 12.